# DNA Methylation as a Future Therapeutic and Diagnostic Target in Rheumatoid Arthritis

**DOI:** 10.3390/cells8090953

**Published:** 2019-08-22

**Authors:** Marzena Ciechomska, Leszek Roszkowski, Wlodzimierz Maslinski

**Affiliations:** 1Department of Pathophysiology and Immunology, National Institute of Geriatrics Rheumatology and Rehabilitation, 02-635 Warsaw, Poland; 2Department of Rheumatology, National Institute of Geriatrics Rheumatology and Rehabilitation, 02-635 Warsaw, Poland

**Keywords:** DNA methylation, epigenetics, biomarkers, rheumatoid arthritis, autoimmunity, therapy

## Abstract

Rheumatoid arthritis (RA) is a long-term autoimmune disease of unknown etiology that leads to progressive joint destruction and ultimately to disability. RA affects as much as 1% of the population worldwide. To date, RA is not a curable disease, and the mechanisms responsible for RA development have not yet been well understood. The development of more effective treatments and improvements in the early diagnosis of RA is direly needed to increase patients’ functional capacity and their quality of life. As opposed to genetic mutation, epigenetic changes, such as DNA methylation, are reversible, making them good therapeutic candidates, modulating the immune response or aggressive synovial fibroblasts (FLS—fibroblast-like synoviocytes) activity when it is necessary. It has been suggested that DNA methylation might contribute to RA development, however, with insufficient and conflicting results. Besides, recent studies have shown that circulating cell-free methylated DNA (ccfDNA) in blood offers a very convenient, non-invasive, and repeatable “liquid biopsy”, thus providing a reliable template for assessing molecular markers of various diseases, including RA. Thus, epigenetic therapies controlling autoimmunity and systemic inflammation may find wider implications for the diagnosis and management of RA. In this review, we highlight current challenges associated with the treatment of RA and other autoimmune diseases and discuss how targeting DNA methylation may improve diagnostic, prognostic, and therapeutic approaches.

## 1. Introduction

Rheumatic diseases are autoimmune disorders characterized primarily by pain and inflammation. Arthritis, which means “joint inflammation”, belongs to lifestyle diseases. In the industrialized world, rheumatic diseases affect more individuals than any other disease group. A third of people of all ages are affected by rheumatic diseases at some point during their lifetime. Rheumatic diseases tend to be chronic and progressive. The economic burden of rheumatic diseases has increased over the last decade. It is estimated that the costs of rheumatic diseases treatment reached a level of 200 billion Euros per year in Europe alone [1]. Some recent studies have revealed that rheumatic diseases are the most expensive of all diseases for European health care systems [2]. There are more than 200 different conditions that are labeled as rheumatic diseases, including rheumatoid arthritis (RA). RA is a long-term autoimmune disease of unknown etiology that affects as much as 1% of the population worldwide [3]. RA is characterized by the presence of autoantibodies to immunoglobulin G (rheumatoid factor—RF) and citrullinated proteins (anti-citrullinated protein antibodies—ACPAs). Although autoantibodies are an important characteristic of RA, some individuals are negative for these autoantibodies. The disease is complex and involves many environmental factors that trigger disease in genetically susceptible individuals [4]. Unfortunately, there is no cure for rheumatic diseases that would completely cure the patient. The most important therapeutic aims for treating RA are eliminating symptoms (such as joint pain, swelling, and stiffness), preventing further joint damage, maximizing physical function, and improving the quality of life. These targets are accomplished by achieving disease remission, a state in which no or only minimal residual inflammation is discernible. Today, this aim is attainable in up to 50% of patients with RA, using approved drugs [5]. Drugs that are currently available are glucocorticoids, nonsteroidal anti-inflammatory drugs or pain medications, synthetic disease-modifying anti-rheumatic drugs (DMARDs), including methotrexate, targeted synthetic DMARDs, which interfere with enzymes, such as Janus kinases (JAKs), biologic DMARDs (Tumor necrosis factor-α (TNF-α) inhibitors, inhibitors of interleukin 6 (IL-6), targets of CD20, inhibitors of CD80/86). These medicines, however, have various side effects that limit their use. Therefore, novel therapies to RA are still needed.

## 2. Molecular Mechanism of RA Development

RA primarily affects synovial joints, which means that the balance between recognition of pathogens and avoidance of self-attack is impaired and the immune system attacks and destroys its healthy tissue [6]. In RA, there is increased recruitment and migration of immune cells from the bloodstream into the target tissue, including synovial membrane or synovial fluid. Consequently, such an influx of activated immune cells producing an enhanced level of pro-inflammatory cytokines leads to the progressive erosion of articular cartilage. Leukocytes, including B cells, T cells, and phagocytes, are the main types of immune cells in the rheumatoid synovium. Indeed, macrophages, along with granulocytes, are responsible for the production of pro-inflammatory cytokines, chemokines, and reactive oxygen species (ROS) that accompanies classical inflammation [7]. Besides, B lymphocytes play critical roles in the pathogenesis of RA. They are the source of RF and ACPAs, which contribute to the immune complex formation and complement activation in the joints. B cells are also very efficient autoantigen-presenting cells and enhance T cell activation through the expression of proinflammatory cytokines [8]. Apart from the production of the proinflammatory cytokines, T cells also activate apoptosis-resistant fibroblast-like synoviocytes (FLS). FLS are the key effector cells in the synovium. These cells are a key component of the hyperplastic, inflamed, and invasive synovial tissue mass that extends from the RA synovium. This results in the degradation of articular cartilage and bone. FLS secrete cytokines that perpetuate inflammation and produce proteases, including matrix metalloproteinases (MMPs), that contribute to cartilage destruction. Rheumatoid FLS develop a unique aggressive phenotype that increases invasiveness into the extracellular matrix (ECM) and further exacerbates joint damage. Indeed, FLS, together with immune cells, create an inflammatory environment in the synovium that ultimately contributes to the joint destruction and overall RA development.

Although the predetermined risk factors, such as genetics, play a role in disease development, genetics does not fully explain the risk of RA development. For instance, it has been shown that monozygotic twins have only 12–15% of concordance rate for RA development, suggesting that environmental and epigenetic changes could strongly account for the non-genomic components of diseases susceptibility [9,10,11]. It has been reported that epigenetic mechanisms mediate the development of chronic inflammation by increased expression of pro-inflammatory cytokines, including IL-1β, TNF-α, IL-6, and ROS induction. These molecules are constitutively produced by a variety of immune cells and FLS under chronic inflammatory conditions that subsequently result in the development of autoimmune rheumatic disorders. Agents that possess both the ability to selectively reduce inflammation resulted in the maintenance of joint integrity and to help manage side effects have great potential in RA treatment.

As opposed to genetic mutation, epigenetic changes are reversible and make them good therapeutic candidates, modulating the immune response or FLS activity when it is necessary. Indeed, the use of epigenetic drugs has been observed in the treatment of various types of cancer, neurological conditions, or heart diseases [12]. Thus, epigenetic therapies controlling autoimmunity and systemic inflammation may find wider implications for diagnosis and management of RA.

## 3. Epigenetic Predisposition for RA Development with Particular Emphasis on DNA Methylation

Epigenetic modifications play a central role in the cellular programming of gene expression. Epigenetics is defined as the study of heritable and reversible changes in gene function that do not involve alterations in the DNA sequence [13]. Epigenetic mechanisms are sensitive to external stimuli; therefore, they mediate in gene-environment interactions. Three main epigenetic mechanisms have been described, including non-coding RNA species, histone modification, and DNA methylation. 

### 3.1. DNA Methylation 

DNA methylation is the most commonly studied epigenetic modification because it poses many advantages over other epigenetic modifications. For example, DNA methylation can be stably inherited through multiple cell divisions. Besides, DNA methylation is the only epigenetic modification that survives the DNA extraction and purification process, even three decades of archival sample storage, as opposed to RNA or proteins [14]. DNA methylation is induced by a family of DNA methyltransferases (DNMTs) (Figure 1). 

DNMT1 is the most abundant DNA methyltransferase and is considered to be the key maintenance methyltransferase in mammals [15]. DNA methylation is a process by which methyl groups (CH3) are added to cytosine at the carbon 5 position (Figure 1). Insertion of a methyl group to DNA sequence results in a modification of chromatin structure and, ultimately, gene silencing (Figure 2). In humans, DNA methylation is almost exclusively found in CpG islands. CpG islands are the regions with a high frequency of Cytosine-phosphate-Guanine dinucleotide (CpG) sites. In the human genome, there are approximately 30,000 CpG islands. Importantly, approximately 60–70% of annotated gene promoters are associated with CpG islands, suggesting that methylation of CpG islands is an important component of gene expression regulation [16]. The relationship between non-mutagenic environmental factors, influencing epigenetics alteration and subsequently pathology, has been examined primarily through the studies of the DNA methylation pattern in animal models [17]. A most notable example is the agouti yellow mice, which are protected from the development of inflammation-mediated diabetes and cancer following a vitamin B12 rich diet (vitamin B12 acts as methyl donors). 

#### 3.1.1. DNA Methylation-Targeted Drugs

Many studies have shown that DNA methylation inhibitors, including 5′-AZA (azacitidine), are already tested in studies aiming to treat various types of cancer and rheumatic diseases. In Table 1, we highlight the main drugs that affect the inhibition or activation of DNA methyltransferases and DNA hydroxymethylation. 

5′-AZA, decitabine, and zebularine belong to the family of nucleoside-derived inhibitors, which were originally tested and approved by the FDA to treat various types of cancer [26,44,45]. Indeed, 5′-AZA is involved in hypomethylation of tumor suppressor genes and subsequently in their activation. Furthermore, nucleoside-derived inhibitors were also tested in treating patients with rheumatic diseases. It was found that the use of 5′-AZA decreased the release of pro-inflammatory cytokines, including IL-6 and TNF-α, by FLS in RA animal model [38]. Fu et al. demonstrated that the promoter of IL-10, which is crucial in inhibition of inflammation, was hypermethylated in four different regions of CpG site in RA patients [19]. This may explain why the level of IL-10 is reduced in RA. Furthermore, stimulation of RA peripheral blood mononuclear cells (PBMCs) with 5′-AZA resulted in elevated production of IL-10. These data suggest that 5′-AZA-mediated-demethylation of IL-10 promoter may inhibit RA development by increased production of immunosuppressive IL-10. We also demonstrated that treatment with 5′-AZA was able to induce strong collagen production and miRNA-135b expression in fibroblasts, suggesting that the demethylation process accelerates fibrogenesis [21]. We also noticed that the level of methyl CpG binding protein 2 (MeCP2) was increased in systemic sclerosis (SSc) fibroblasts compared to healthy control (HC). MeCP2 is capable of binding specifically to methylated DNA and regulate gene silencing. Surprisingly, dermal fibroblasts treated with Transforming growth factor-β (TGF-β) increased the production of MeCP2, suggesting that profibrotic TGF-β is involved in the acceleration of DNA methylation [21]. 5′-AZA treatment also increased the level of single-stranded DNA-binding protein (SSB) in a human salivary gland cell line, mimicking global DNA demethylation status in patients with primary Sjogren’s syndrome (pSS) [20]. Importantly, antibodies against SSB/La are commonly found in pSS patients. It was found that expressions of Tumor-necrosis factor receptor-associated factor 1 (TRAF1) and connective tissue growth factor (CTGF) were significantly increased in osteoarthritis (OA) and RA chondrocytes following decitabine treatment [23]. Besides, it has been demonstrated that zebularine has synergistic activity with interferon-γ (IFN-γ) in the induction of immunosuppressive enzyme indoleamine 2,3-dioxygenase-1 (IDO1). Therefore, zebularine may be an important immunosuppressive agent in chronic autoimmune diseases treatment [25]. All these results suggest that nucleoside-derived inhibitors can be used to modulate RA and other chronic inflammatory conditions. Unfortunately, 5-AZA is unstable in an aqueous solution and is highly toxic to cells. This results in serious side effects, including nausea, fatigue, neutropenia, thrombocytopenia, and reduced sperm function, in patients treated with 5′-AZA [46]. Therefore, some studies have started to use the next-generation of DNMTs inhibitors, which are less toxic, including RG108. RG108 is a non-cytidine analog (in contrary to 5′-AZA, decitabine, and zebularine) which has been successfully tested in human prostate cancer cells [27], breast cancer cells [28], or in human colon cancer cells [29,47] but not in RA. Importantly, RG108 does not direct demethylation of minor satellite repeats, indicating that the RG108 administration will not produce cellular genomic instability and potential mutagenesis. Other drugs assessed for their potential to induce hypomethylation in solid tumors are hydralazine (vasodilator) and procainamide (antiarrhythmic agent) [48]. Hydralazine has been reported to prevent the activity of DNMTs by the interaction of its nitrogen atoms with the Lys-162 and Arg-240 residues of the enzyme, whereas procainamide acts similarly as a competitive inhibitor by preferentially binding to DNMT1 [38,49]. Lately, the lipophilic, quinoline-based compound SGI-1027 was demonstrated to be a novel DNMT inhibitor in vitro. SGI-1027 causes the degradation of DNMT1 and demethylation of the cyclin-dependent kinase inhibitor 2A (CDKN2A), MutL homolog 1 (MLH1), and tissue inhibitor matrix metalloproteinase-3 (TIMP3) genes [50]. Another strategy to inhibit DNMT1 includes the use of antisense oligonucleotides and miRNAs. MG98 is a short-chain oligodeoxynucleotide that specifically binds to the 3′ untranslated region (3′UTR) of DNMT1 mRNA to stop its translation. The multicenter study demonstrated that combination of MG98 and type I IFN treatment resulted in partial response or stable disease in some metastatic renal cell carcinoma patients, suggesting the safety and clinical activity of MG98 [51]. Data also suggest that epigallocatechin-3-gallate (EGCG) can restore or reactivate the expression of the DNA hypermethylation-silenced genes in human skin cancer cells by downregulation of DNMT and histone deacetylase (HDAC) activity. However, unlike 5-AZA, EGCG has a double-action involving both DNA demethylation and modifications of histone acetylation and methylation [37]. Several studies show that procaine (anesthetic drug) is capable of reducing the methylation level of CpG island in some cancers, such as hepatoma, breast cancer, and gastric carcinoma, through inhibiting the binding of DNMT1 to the target genes [31,49,52]. On the other hand, DNA is hypomethylated in an animal model of lung cancer compared to normal tissue. Treatment with budesonide, which is DNA methyltransferase activator, resulted in increased methylation of DNA and ultimately decrease in tumor size [40,41]. This suggests that modification of DNA methylation either by hypo- or hypermethylation is effective in different types of tumors. Budesonide is not only the DNA methyltransferase activator but also acts as glucocorticosteroid to treat asthma, and it has been shown to improve tender joint and swollen joint counts in RA patients [40]. 

#### 3.1.2. DNA Methylation Pattern in RA

DNA methylation pattern is also altered in RA, affecting immune response and FLS activity, which consequently results in disease development. Indeed, global DNA hypomethylation is demonstrated in RA peripheral blood mononuclear cells (PBMCs) and RA FLS [53]. Nakano et al. demonstrated that 1859 genes relevant to cell movement, adhesion, and trafficking were differentially methylated in RA FLS compared to FLS derived from OA, suggesting that different methylation signatures could alter gene expression in FLS and contribute to the pathogenesis of RA [54]. The hypomethylated loci are identified in key genes relevant to RA, such as Signal transducer and activator of transcription 3 (STAT3). Constitutive activation of STAT3 correlates with an increased level of proinflammatory IL-6, which is known to play a pivotal role in chronic inflammation in autoimmune diseases, including RA [55]. Surprisingly, the activity of DNMTs was similar in RA and OA FLS. Stimulation with proinflammatory IL-1β or TNF-α reduced DNMTs functional activity. Another study demonstrated that there were differences in methylation of IL-6 between RA knee and hip FLS, indicating that IL-6-related mechanisms might vary from joint to joint [56]. Recent studies have identified a novel methylation signature in T and B cells of patients with early RA compared to healthy control (HC) [57,58]. Besides, a subset of sites, which are differentially methylated in early RA, displayed similar changes to those seen in established disease. This suggests that changes in DNA methylation pattern, which occurred during the early stages, play a role in RA development. Also, the methylation levels of IL-6 promoter in PBMCs are significantly lower in RA patients than those in HC [59]. It has been shown that the promoter region of C-X-C Motif Chemokine Ligand 12 (CXCL12) is hypomethylated and results in increased MMPs expression and joint destruction in RA patients [60]. In contrast, the gene coding for dual-specificity phosphatase 22 (DUSP22) is hypermethylated in T cells in RA [61]. DUSP22 is a tyrosine phosphatase which negatively regulates the IL-6 transcription factor STAT3. Besides, hypomethylation of Death receptor 3 (DR3) results in its increased expression in RA FLS. These findings may explain the resistance to apoptosis of synovial cells in RA [62]. Some studies have shown that long-term exposure to the proinflammatory cytokine IL-1 can contribute to the RA FLS DNA methylation pattern through altered DNMT expression. Only some loci are affected by IL-1, suggesting that other inflammatory mediators are involved or that the cells are irreversibly imprinted through other mechanisms or before the onset of arthritis [63]. 

Another study has shown that DNMTs can promote and stabilize Forkhead Box P3 (FOXP3) expression in anti-inflammatory Treg cells. Indeed, mice with conditional deletion of Dnmt1 in their Treg population develop autoimmunity and subsequently dies by 3 to 4 weeks earlier due to increased inflammatory genes expression, widespread mononuclear cell tissue infiltration, and exuberant T and B cell responses [64]. Adoptive transfer of wild type Treg prevents autoimmunity in mice with Dnmt1^−^/^−^Foxp3+ Treg. This implies that Dnmt1 is necessary for the maintenance of the core gene program underlying Treg development and function. The mechanism of activation or inhibition of DNA methylation by selected drugs is demonstrated in Figure 2. Another study found commonalities across various autoimmune diseases, including Graves’ disease (GD), RA, systemic lupus erythematosus (SLE), and SSc [65]. Multiple type I IFN-related genes are related to the pathogenesis of one or more of the autoimmune diseases mentioned above from genetic association studies, including STAT4, IRF5, IFIH1, and PLZF [66,67,68]. The similarity of methylation profiles across these diseases and the common hypomethylation signature of type I IFN-related genes in CD4+T cells have also been identified [69].

In contrast to most cancers, in which hypermethylation is a widespread character, the genes in RA tend to be hypomethylated [70]. Fang et al. demonstrated that four genes, i.e., Fc receptor-like A (FCRLA), Coiled-Coil Domain Containing 88C (CCDC88C), B-cell lymphoma 11B (BCL11B), and Apolipoprotein L6 (APOL6), are hypomethylated and down-regulated in RA compared with OA samples. Reduced expression of hypomethylated CCDC88C influences Wnt signaling pathway and subsequently promotes RA progression. Fang et al. also found that APOL6 is an important apoptosis-related protein, which is critical for the progression of RA; therefore, APOL6 may be a novel biomarker in RA [71]. In another study, it was found that hypomethylated regions in the Cytochrome P450 2E1 (CYP2E1) and DUSP22 gene promoters were associated with the active and erosive disease, respectively. Pathway analyses suggest that biological mechanisms underlying each clinical outcome are cell-type specific [61]. Evidence for independent effects on DNA methylation from smoking and medication use were also identified; however, no regions of differential methylation were significantly associated with smoking and DMARD treatment [61]. Furthermore, an epigenome-wide study of monozygotic twins discordant for RA found that DMARD treatment led to hypomethylation of the promoter of Ring finger protein 5 (RNF5) and 1-acyl-sn-glycerol-3-phosphate acyltransferase alpha (AGPAT1) genes that have been implicated in inflammation and autoimmunity [72]. But the study did not specify which DMARDs were prescribed to the patients. One of the studies has investigated DNA methylation as a predictive biomarker for response to anti-TNF biologic drugs with success. Using an epigenome-wide association study in whole blood, which compared DNA methylation patterns in good and non-responders to etanercept (TNF-α inhibitor), two significantly differentially methylated positions mapping to Low-density lipoprotein receptor-related protein 1 (LRPAP1) were identified [73]. 

#### 3.1.3. DNA Hydroxymethylation 

In addition to DNA methylation, we can also observe DNA hydroxymethylation, which is an epigenetic modification mediated by Ten-Eleven Translocation (TET) family proteins [74]. TET enzymes are responsible for the oxidation of 5-methylcytosine (5mC) into 5-hydroxymethylcytosine (5hmC), which subsequently leads to DNA demethylation. Increased expression of TET1-TET3 enzymes in monocytes and TET2 in T cells has been shown to lead to abnormal global DNA hydroxymethylation in early RA patients [75]. Importantly, methotrexate treatment partly reduces the level of DNA hydroxymethylation. Zhang et al. observed high levels of TET1, TET2, and 5hmC, probably generated by the two previous enzymes, in CD4+ T cells from patients with SLE compared to HC. Therefore, hydroxymethylation of DNA was proposed as a potential mechanism, leading to hypomethylation of DNA in lupus [76]. The study also showed that genome-wide hydroxymethylation occurred during an experimental autoimmune encephalomyelitis induction and TET proteins acted as important regulators in this process. DNA hydroxymethylation has been used in various studies in multiple sclerosis [76]. TET inhibition induced by AGI-5198 and HMS-101 inhibitor compound leads to growth suppression, respectively, of glioma cells and acute myeloid leukemia (AML) [42,43]. These data suggest the potential therapeutic use of TET inhibitors in cancer, demyelinating diseases, and also in RA. Overall, the aforementioned examples indicate that the DNA methylation landscape is profoundly impaired in RA and other rheumatic diseases. 

### 3.2. Non-Coding RNA 

Non-coding RNA include microRNAs (miRNAs) and long noncoding-RNAs, whose function is still being investigated. MiRNAs are endogenously encoded single-stranded RNAs that post-transcriptionally regulate gene expression by targeting and degrading mRNA transcripts. It has been shown that miRNAs play an important role in the progression of rheumatic diseases [77]. Our previously published data showed that miRNA-5196, miRNA-29a, and miRNA-135b targeting profibrotic Fos-related antigen 2 (Fra2), Tissue inhibitor matrix metalloproteinase-1 (TIMP-1), and Signal transducer and activator of transcription 6 (STAT6), respectively, were able to reverse the profibrotic properties of SSc fibroblasts [21,78,79]. In RA patients, the level of miRNA-146a is significantly elevated in CD4+ T cell subset and positively correlates with TNF-α concentration [80,81]. Also, the level of miRNA-146a and miRNA-150 is elevated in IL-17 producing T cells [82]. In contrast, Zhu et al. reported that miRNA-23b inhibited IL-17-associated inflammation by targeting TGF- β binding protein 2 (TAB2) and TAB3 [18]. Our recent results demonstrated that sera circulating miRNA-5196 could be used as a biomarker predicting positive treatment outcome of TNF-α therapy in RA and ankylosing spondylitis patients, whereas single-nucleotide polymorphisms (SNPs) of miRNA-146a contribute to RA development [2,83]. Similarly, using genome-wide association study (GWAS) data, Wohlers et al. reported that SNP rs968567 affected the expression of miR-1908-5p and contributed to RA predisposition [84]. Besides, enhanced expression of miRNA-203 in RA FLS is associated with hypomethylation of MMP1 and IL-6 gene promoters [85]. This may suggest a potential link between miRNA and DNA methylation in RA pathogenesis. 

### 3.3. Histone Modifications

It has been also found that alteration in histone modification, which is another epigenetic modification, can contribute to RA development [86]. Histone modification is a covalent post-translational modification to histone proteins which includes methylation, phosphorylation, acetylation, ubiquitylation, and sumoylation of histone N-terminal tail domains and also core domains [12]. The balance of histone acetylases (HATs) vs. histone deacetylases (HDACs) is strongly shifted toward chronic histone hyperacetylation in RA patients [87]. Indeed, the level of histone H3 acetylation in the IL-6 promoter is significantly higher in RA FLS compared to OA FLS [88]. In contrast, treatment with curcumin, which is an inhibitor of HAT, leads to reduced IL-6 secretion. These data imply that epigenetic mechanisms are important in targeting RA pathogenesis. 

HDAC inhibitors have been studied in the context of cancer therapy and can be considered for their use in RA, due to the aggressive behavior of FLS. One of the most frequently used HDAC inhibitors to target FLS is trichostatin A (TSA) [89]. Treatment of RA FLS with TSA significantly increases apoptosis but decreases the inflammatory response and invasiveness [90,91,92,93,94]. Overall, it appears that histone acetylation patterns are altered in FLS, and inhibition of HDACs activity reduces inflammation and joint damage in preclinical models. Although each HDAC can control different genes and functions in FLS, and HDAC inhibitors often target multiple HDACs, HDAC inhibition could be a promising option for the treatment of RA. We have also shown that specific histones modification induced by histone methyltransferase inhibitor 3-Deazaneplanocin A (DZNep), but not by apicidine (histone deacetylase inhibitor) can induce strong TIMP-1 production in SSc monocytes and promotes pathogenic properties of myofibroblasts [95]. Similarly to RA, close crosstalk between histone modification and DNA methylation in MMP-9 induction has been demonstrated in diabetes [96]. Previous studies also showed that enhanced level of Methyl-CpG-binding protein 2 (MECP2) played an important role in RA fibroblasts and linked two epigenetic repression mechanisms, including DNA methylation and histone deacetylation [97]. Indeed, MECP2 can selectively bind the methylated DNA and to interact with HDAC-molecules and it promotes canonical Wnt pathway in RA fibroblasts [97]. In contrast, MECP2 has protective, anti-fibrotic effects in SSc fibroblasts [98]. Furthermore, SNPs in the MECP2 gene have been implicated with juvenile idiopathic arthritis (JIA) and SLE predisposition [99,100]. Many research reported that MECP2 is an important oncogene with a strong epigenetic mode of action. Indeed, MECP2 can increase the activity of pro-oncogenic Ras or Wnt pathways and suppresses apoptosis in many types of cancer [101,102,103]. This suggests that MECP2 activates a similar signaling pathway in cancer cells as in auto-aggressive RA FLS. Therefore, drugs modulating MECP2 expression represent a promising target not only for treating cancer but also for treating RA. 

## 4. Cell-Free Methylated DNA (ccfDNA) as Biomarkers

An increasing amount of evidence has demonstrated that early diagnosis, efficient treatment initiation, and early achievement of remission are the major predictors of long-term clinical and functional outcomes. Indeed, controlling disease activity in a very early window of opportunity offers unique sustained benefits [104]. It has been shown that early and aggressive RA treatment within 12 weeks significantly improves patients’ outcomes by 30% pain reduction compared to patients who received treatment after 12 weeks [105]. Importantly, better patients’ stratification in terms of susceptibility to rapid joint destruction and initiation of a more aggressive treatment in these patients may prevent irreversible joint damage. This approach may not only delay or reduce disease progression but also effectively decrease the cost of health care, hospital admission, and social service. It is, therefore, essential to identify the biomarker signature that could predict joint damage at an early stage and would support more informed clinical decisions on the most appropriate treatment regimens for individual patients. Recent studies have shown that circulating cell-free methylated DNA (ccfDNA) in blood offers a very convenient, non-invasive, and repeatable “liquid biopsy”, thus providing a reliable template for assessing molecular markers of various diseases. Most of these studies have been done in cancer. Indeed, it has been shown that tumor ccfDNA, which is detectable in the bloodstream, contains the same molecular aberration as the solid tumors. Therefore, sampling ccfDNA via blood overcomes the problem related to accessibility and subsequently painful biopsies. Aberration in the DNA methylation pattern is a common feature of different types of cancer, including breast, prostate, liver, or lung cancer. Importantly, these changes occur early in cancer development and typically represent the expression of tumor suppressor genes since increased methylation corresponds to gene inhibition. 

DNA methylation changes have been reported to occur early in the carcinogenic process and are potentially good indicators of an early disease [106]. For example, methylated Glutathione S-transferase Pi 1 (GSTP1) is observed in more than 90% of patient’s sera as an early marker of resistance to treatment in prostate cancer [107]. Importantly, the level of methylated GSTP1 in plasma is a better predictor of the overall survival than prostate-specific antigen (PSA), which is used in most common prostate cancer tests. It has been also reported that Stratifin (SFN) is methylated in 96% of breast cancer patients [107,108], whereas Homeobox A9 (HOXA9) and Engrailed homeobox 1 (EN1) are methylated in 95% and 80% of serum patients with ovarian tumor, respectively [109]. Besides, the amount of shed ccfDNA in the blood is positively correlated with tumor size [110]. This implies that cancer-specific ccfDNA can be used to quantify tumor DNA, provide prognostic information regarding tumor burden, as well as reveal methylation patterns of the tumor. Also, sera isolated ccfDNA of Cadherin 1 (CDH1), DNMT3b, Ethylene response sensor 1 (ERS1) genes display significantly higher diagnostic value than Alpha-fetoprotein (AFP). AFP is a marker and is widely used to monitor people with liver diseases and to discriminate hepatocellular carcinoma from chronic hepatitis B and liver cirrhosis [111]. Apart from examples in cancer, some studies have shown that ccfDNA can be used as non-invasive prenatal testing for Down syndrome (ERG gene) [65] or could be used in non-invasive prenatal diagnostics (RASSF1A gene) [112]. Overall, these results have shown that ccfDNA have great advantages over currently used screening tests for better detection and monitoring of treatment response in cancer and even in prenatal testing. Unfortunately, to the best of our knowledge, no studies have shown clinical application of ccfDNA in RA diagnosis; therefore, it will be of great interest to validate whether sera ccfDNA reflecting changes in joint destruction could help in early diagnosis and subsequently prevention of RA development. 

Currently, RA is primarily diagnosed by visual examination. Blood testing is used as a secondary method for comprehensive analysis. In 2010, new RA classification criteria were jointly published by the American College of Rheumatology and the European League Against Rheumatism. These criteria consist of four domains, including joint involvement, serologic abnormality based on RFs and ACPAs tests, elevated acute phase response (ESR and CRP), and the duration of arthritis [113]. Currently, these commonly used serologic parameters (RFs and ACPAs) as specific biomarkers constitute only one-third of the total points. Whereas, 50% of the points are from joint involvement and are analyzed by subjective and error-prone visual examination. Measurement of RF is often negative with early RA; therefore, the specificity of the RF test has been limited [114]. Similarly, the sensitivity of the ACPAs test is lower for RA diagnosis compared to other autoimmune diseases. For instance, ACPAs antibodies are positive before diagnosis in only 33.7% of the RA patients [115]. This indicates that new RA classification criteria are also not perfect, and there is no specific and reproducible serum biomarker for RA diagnosis. Attempts to learn about new biomarkers have been going on for years. Biomarkers help rheumatologists to judge a group of patients that can improve the diagnosis and prognosis, and further facilitate appropriate and precise treatment with targeted therapy. Here are some examples of currently used biomarkers: acute phase (ESR, CRP, ferritin, serum amyloid A, procalcitonin), antibody (IgG, RF, IgG-RF, ACPAs), bone metabolism and pathogenesis-related (MMP-3, sICAM-1, CXCL-13, KL-6, IL-6, LRG) [116]. It should be noted that there may be other satisfactory biomarkers of response. There is growing evidence that implicates dysregulation of miRNAs in blood, T cells, and synovial fibroblasts in inflammation and joint destruction [117]. Due to the insufficient availability, sensitivity, and specificity of these biomarkers, further work is needed. Although MRI techniques have been developed for early-stage evaluation of cartilage damage in RA, these techniques require expensive instrumentation facilities as well as being contraindicated in people who have implanted devices, such as pacemakers or aneurysm coils. Thus, evaluation of new serological biomarker, including ccfDNA, with high medical utility is important to improve the diagnosis of RA. Ideally, such biochemical test should be minimally invasive, relatively cheap, and be able to detect and distinguish common types of arthritis at the early stage. 

## 5. Conclusions 

In this review, we highlight the current challenges associated with the diagnosis and treatment of RA and other autoimmune diseases. Unfortunately, studies on the clinical use of epigenetic drugs modulating aberrant DNA methylation pattern in RA are at a very early stage. Much more research can be observed in studies on DNA methylation in the treatment and diagnosis of cancer and proliferative diseases. Indeed, the FDA already approved blood tests based on DNA methylation biomarker technology to screen colorectal cancer [118] and soon is expected to be approved to detect lung cancer. Therefore, we hope that, in the near future, we will be able not only to better understand how epigenetic landscape contribute to the pathogenesis of RA but also to use such DNA methylation mechanisms in RA treatment and diagnosis. The overarching goal is to find new non-invasive DNA methylation biomarkers that can be used in everyday practice to detect early RA before irreversible joint destruction may occur. This may not only delay or reduce disease progression but overall decrease the costs of health care.

## Figures and Tables

**Figure 1 cells-08-00953-f001:**
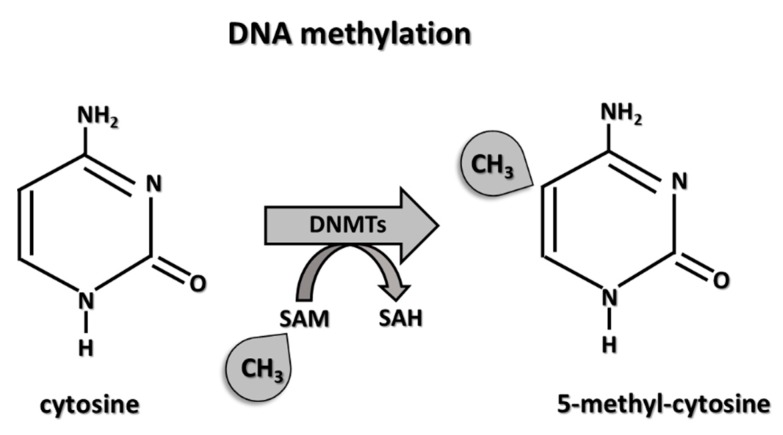
The mechanism of DNA methylation. DNA methylation is exerted by DNA methyltransferases (DNMTs). DNMTs at the 5′-position of cytosine residues in CpG dinucleotides transfers methyl groups from SAM (S-adenosylmethionine) to SAH (S-adenosylhomocysteine); thus, 5-methylcytosine is formed.

**Figure 2 cells-08-00953-f002:**
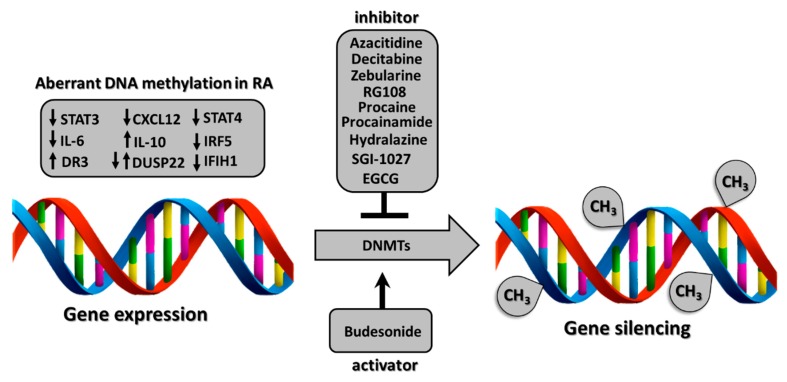
Potential mechanism of differentially methylated genes in rheumatoid arthritis (RA) regulated by various epigenetic drugs. DNA methylation is exerted by DNMTs. DNMTs are influenced by various epigenetic drugs either inhibiting or activating DNMTs, which consequently results in modulation of gene expression. Down arrow indicates hypomethylated genes (*STA3*, *IL-6*, *CXCL12*, *DUSP22*, *STAT4*, *IRF5*, *IFIH1*), and up arrow indicates hypermethylated genes (*DR3*, *IL-10*, *DUSP22*) in RA.

**Table 1 cells-08-00953-t001:** Various drugs, inhibiting or activating DNA methylation and DNA hydroxymethylation, which are demonstrated in different diseases.

Drug	Epigenetic Effect	Disease
Azacitidine (5′-AZA)	DNA methyltransferase inhibitor	RA [18,19], primary Sjogren’s syndrome (pSS) [20], systemic sclerosis (SSc) [21], myelodysplastic syndromes (MDS) [22]
Decitabine	DNA methyltransferase inhibitor	RA [23], OA [23], chronic myelomonocytic leukemia [24]
Zebularine	DNA methyltransferase inhibitor	autoimmunity/chronic inflammation [25], MDS [26]
RG108	DNA methyltransferase inhibitor	prostate cancer [27], breast cancer [28], colon cancer [29]
Procaine	DNA methyltransferase inhibitor	SS [30], gastric carcinoma [31]
Procainamide	DNA methyltransferase inhibitor	RA [32], SLE [32], drug-induced autoimmunity [33], solid tumors [34]
Epigallocatechin-3-gallate (EGCG)	DNA methyltransferase inhibitor	RA [35,36], skin cancer [37]
Hydralazine	DNA methyltransferase inhibitor	RA [32], SLE [32], cervical cancer [38]
SGI-1027	DNA methyltransferase inhibitor	hepatoma [39]
Budesonide	DNA methyltransferase activator	lung cancer [40,41]
AGI-5198	DNA hydroxymethylation inhibitor	brain tumor [42]
HMS-101	DNA hydroxymethylation inhibitor	acute myeloid leukemia (AML) [43]

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
