# Peer review of "DNA Methylation as a Future Therapeutic and Diagnostic Target in Rheumatoid Arthritis"

_cells, 2019, doi:10.3390/cells8090953_

Round 1
Reviewer 1 Report
In this well written paper, M Ciechomska and colleagues have presented an up-to-date review regarding the emerging interest to use DNA methylation as a future therapeutic and diagnostic target in rheumatoid arthritis:
I have some minor comments:
DNA methylation biomarkers highlighted in RA (lines 359-380) have not been generated from the analysis of cell free methylated DNA, which is confusing for the reader. The cross-talk between DNA methylation/histone modifications and gene polymorphism in RA is lacking.
Author Response
Thank you for the comments. We have moved lines 359-380 to section “DNA methylation pattern in RA”. We have addressed the cross-talk between DNA methylation/histone modifications and gene polymorphism in RA in sections “3.2 Non-coding RNA” and “3.3 Histone modifications”.
Reviewer 2 Report
Manuscript entitled „DNA methylation as a future therapeutic and diagnostic target in rheumatoid arthritis” submitted to Cells is a well written review paper on new possibilities for treatment and diagnosis of rheumatoid arthritis with the use of epigenetics. Basically I have no serious comments and recommend the paper to be published. The only issue that could be considered to correct is the structure of the article – I would place the paragraph „DNA methylation before „Non-coding RNA” and „Histone modifications”.
Author Response
Thank you for the comments. We have addressed this comment and moved “3.1 DNA methylation” section before „3.2 Non-coding RNA” and „3.3 Histone modifications”.
Reviewer 3 Report
The review article titled “DNA methylation as a future therapeutic and diagnostic target in rheumatoid arthritis” nicely summarizes the epigenetic landscape of rheumatoid arthritis (RA), especially DNA methylation in rheumatoid arthritis (RA). The article is well-organized and brings forth important developments in the field.
However, the review could be improved by incorporating the following suggestions:
Minor comments: Please check the sentences and use consistent grammar throughout the article. A few instances which need revision are indicated: Please check and correct line no.105 to 109, line no. 132 to line no.133, line no 266 to line no.268, line no.272 to line no 274, line no.280 to line no. 281. Some of the statements in the article need to be referenced against the original source. For e.g: Line no.106 to Line no 108, should have the appropriate reference. It would be useful to use a more recent reference to indicate the efficacy of MG98 in clinical settings. In the article, the source of the information regarding the status MG98 in clinical settings is from 2002. However, it will be relevant for the readers to have updated information on the status of clinical trials.
Major Comments: Section 3 Line no.104 and Line no. 131: The epigenetic mechanisms of non-coding RNA and histone modifications have been described by several previous review articles focussed on the description of the global epigenetic landscape in diseases. A recent case in point being: Mazzone, R., et al. (2019). "The emerging role of epigenetics in human autoimmune disorders." Clinical Epigenetics 11(1): 34. The present review does not need to reiterate information available to the readers and may benefit by focusing specifically on DNA methylation in RA. In section 4 of the review, under the heading of "Cell free methylated DNA as biomarkers", the relevance of inclusion of Line no 359 to Line no. 380 discussing the hypomethylation of genes in RA, is not clear. The section could be improved by clearly outlining how Cell free methylated DNA have been used as biomarkers, without including details on the specific details of hypomethylation in RA. Please discuss the mechanistic similarities between cancer and RA from a immunological perspective. Please discuss how certain anti-cancer epigenetic regulators could be effective in RA, considering RA is primarily an autoimmune disorder.
Author Response
Thank you for the comments.
Regarding Minor comments: We have addressed the comment regarding lines 105-109, 132-133, 266-268, 272-274, 280-281 by changing the sentences or by adding the original references. We have also added updated information regarding MG98 in clinical study.
Regarding Major comments We have addressed the comment by reducing the sections “3.2 Non-coding RNA” and “3.3 Histone modifications”. We have moved lines 359-380 to section “DNA methylation pattern in RA”. We have also discussed potential role of anti-cancer epigenetic regulators in RA.
Round 2
Reviewer 3 Report
The authors fairly addressed my previous comments.